# Quality Assurance of Potential Radioanalytical Methods for $^{14}$C in Environmental Samples

Saroa Rozas *, Raquel Idoeta, María Teresa Rodríguez and Margarita Herranz

Department of Energy Engineering, Faculty of Engineering in Bilbao, University of the Basque Country (UPV/EHU), Plaza Ingeniero Torres Quevedo 1, 48013 Bilbao, Spain
* Correspondence: saroa.rozas@ehu.eus; Tel.: +34-94-601-7204

**Abstract:** Today, the measurement of $^{14}$C in environmental samples is of particular interest, as it enables the assessment of the impact caused by nuclear activities and the fossil fuel industry on the environment. In order to assure the quality of $^{14}$C measurement results, the strategy to enlarge the validation of three radioanalytical methods in environmental samples using liquid scintillation spectrometry—the direct counting of water, bubbling of water and combustion of solids—is presented. Due certain difficulties, such as the lack of quality control materials and the scarcity of proficiency test and intercomparison exercises, especially in solid samples, a set of water and soil samples were prepared for the purpose by tracing them with known quantities of a $^{14}$C standard solution at two activity levels. Aliquots were subjected to the corresponding method and their activity concentration was calculated. Finally, uncertainty, detection limit, accuracy, precision, repeatability and linearity were analysed. The acceptance criteria for the quality parameters were previously established according to ISO 13528:2015 standard and Eurachem Laboratory Guide to Method Validation. In all the methods, the studied parameters fall within the acceptance range, so they are validated. The quality of the results in real samples is controlled through field validation.

**Keywords:** radiocarbon; liquid scintillation spectrometry (LSS); quality assurance; validation; environmental assessment

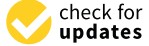



## 1. Introduction

$^{14}$C is a common radionuclide in our environment, due to its long half-life (5730 years) [1], its natural origin in the atmosphere from the reaction between $^{14}$N and cosmic neutrons and its artificial origin from nuclear explosions and releases from the nuclear industry and medical and research activities [2].

As $^{14}$C is a long-lived radionuclide, it remains in the environment for a long time and is integrated in the carbon cycle, mainly as $CO_2$. There, it can be transformed into organic carbon via the photosynthesis of plants and other organisms, such as cyanobacteria, and then into $CO_2$ via breathing or into inorganic carbon through the decomposition of bacteria and fungi. $CO_2$ can also be absorbed and released by oceans, where it is transformed into carbonic acid [3,4].

In both terrestrial and marine ecosystems, natural $^{14}$C tends to reach a stable ratio to the total carbon in different environmental compartments: air, water, soil, sediments, food and biota [2].

Therefore, its measurement is very useful for the assessment of the impact caused by nuclear activities on abiotic and biotic compartments, as they increase their $^{14}$C content and the natural ratio due to $^{14}$C releases [5,6]. It is also very useful to assess the impact of the fossil fuel industry, which may cause the so-called Suess effect [7], i.e., the decrease in the $^{14}$C to the total carbon natural ratio, as it releases carbon with low content of $^{14}$C into the atmosphere.

In order to control the level of $^{14}$C in air, water, soil, sediments, food and biota, the Laboratory of Low Activity Measurements (LMBA) of the University of the Basque Country

(UPV/EHU), has used different methods to analyse this radionuclide for around 15 years. Most of them have been carried out within the framework of the Spanish National Radiological Surveillance Network for the environmental radiological monitoring of nuclear power plants and nuclear fuel cycle facilities.

As $^{14}$C is a medium-energy ($E_{max}$ = 156.476 keV) beta emitter [1], it is measured via liquid scintillation spectrometry (LSS). This radiometric technique allows the result of the $^{14}$C measurement to be expressed in activity unit (Bq), but it is usually expressed in activity concentration units (Bq kg$^{-1}$ or Bq m$^{-3}$).

Even if the radiometric technique is always LSS, the prior sample preparation depends on the type of sample and the detection limit required.

Aqueous matrices are usually analysed following the ISO (International Organization for Standardization) 13162 standard [8], namely, via direct counting or bubbling and $CO_2$ absorption. Direct counting provides good detection efficiencies, the measurement of total $^{14}$C and rapid and inexpensive results. Therefore, it is very useful for screening purposes. However, it is limited by spectral interferers; by variable quench, depending on water purity; and by the maximum volume of sample to be measured, which leads to detection limits higher than 2 Bq L$^{-1}$.

The bubbling and $CO_2$ absorption method also enables the measurement of total $^{14}$C. As it is carbon-selective, there are no spectral interferences during the sample measurement. Moreover, as it supports water samples greater than 50 mL, detection limits become lower than 2 Bq L$^{-1}$. However, the method is limited by the maximum quantity of the $CO_2$ absorber, which is miscible with the scintillation cocktail, and it is strongly affected by its quench and chemiluminescence [9], which lead to low overall efficiencies (around 30%).

Solid samples can be treated with a sample oxidizer, which also allows the determining of the total $^{14}$C [10]. Its use is limited by the maximum quantity of sample that can be put into the sample oxidizer [10], which leads to high detection limits, and also the $CO_2$ absorber, which is a quench agent [9], leading to low overall efficiencies (around 40%), and in some cases, spectral interferences.

The LMBA has implemented a quality system in accordance with the ISO/IEC (International Electrotechnical Commission) 17025 [11], in order to assure the quality of its results, involving analytical procedures, such as $^{14}$C activity determination, in any kind of sample. Therefore, analytical procedures should be validated and controlled, subjecting them to proficiency tests and interlaboratory comparisons using reference materials. This has permitted the laboratory to be accredited by ENAC, the Spanish national accreditation body, under the ISO/IEC 17025 standard [11].

Today, there are many established validation methods for analytical methods, but not so many for the radioanalytical ones. Hence, the validation of radioanalytical methods still pose a challenge for many laboratories, which usually base it on those for analytical methods.

The fundamental items that make the difference between analytical and radioanalytical validation methods are:

- Radioanalytical methods are usually time-consuming (some of them may involve several days for sample preparation).
- Radioanalytical methods validation involves the use of radioactive substances.

Due to the impact that radioactive substances may cause on humans and the environment, most laboratories avoid preparing radioactive materials and validate their procedures externally—taking part in proficiency tests and interlaboratory comparison exercises or using certified reference materials—covering a wide range of activities.

However, in the case of $^{14}$C, validations still pose a challenge for laboratories due to the scarcity of intercomparison exercises, especially in solid samples. In the last 10 years, only two intercomparisons in solids have been located, both in biological (leaves) samples and in the same activity concentration range (100–1000 Bq kg$^{-1}$, dry weight) (IRSN (Institut de Radioprotection et de Sûrete Nucléaire), 2018 and CSN (Spanish Nuclear Safety Council), 2014). We participated in the second one and the result was "accepted" (Table 1). Other intercomparisons in solids, in which we participated, did not yield conclusive results; either

because our results were below the detection limit (NPL (National Physical Laboratory), 2008, and INSIDER (Improved Nuclear SIte characterisation for waste minimisation in Decommissioning and Dismantling operations under constrained EnviRonment), 2020) or because the exercise could not be evaluated due to the lack of a certified reference value and the small number of participating laboratories (CSN, 2010, in diet ashes and CSN, 2018, in milk power).

The accepted exercise only partially serves to validate our method, a validation that was able to be completed using reference materials. The IAEA (International Atomic Energy Agency) provides the C reference material series, whose $^{14}$C values are given in pMC (percent modern carbon); units that do not correspond to the radiometric units used in the radioactivity measurement laboratories (Bq m$^{-3}$ or Bq kg$^{-1}$). It is possible to theoretically transform pMC data into Bq m$^{-3}$ data, but this implies the assumption of certain hypotheses that may cause the measured activity concentration value to not fit with the calculated one.

To overcome this problem, an intercomparison exercise to determine $^{14}$C activity concentrations in three of the nine IAEA reference materials, C2, C6 and C7, was conducted between 10 laboratories in the UK (United Kingdom). The laboratories performed the determinations using a number of techniques. The results obtained were very scattered, and those laboratories using radiometric methods provided results below the detection limit on several occasions, such as samples C2 and C7, which were those with the lowest pMC values [12]. Sample C6, the sample with the highest pMC value and the best radiometrically characterised sample, is no longer available as reference material from the IAEA. So, we performed different determinations on the sample IAEA-C7. The results are presented in Table 1 and can be considered as approved, although the reference value was theoretically calculated from its pMC value.

In the case of aqueous samples, it is easier to find intercomparisons. For example, NPL, IARMA (International Atomic Reference Material Agency), IRSN and CSN organise intercomparisons on aqueous matrices with some regularity. In the last ten years, we have participated in two of them (CSN 2015 and CSN 2019); in both cases, the evaluation was positive (see Table 1).

**Table 1.** $^{14}$C activity concentration (*a*), in Bq kg$^{-1}$, in solid (biological) and water samples from different CSN (Spanish Nuclear Safety Council) proficiency tests (PT) and IAEA (International Atomic Energy Agency) reference material (Ref. Mat.). Uncertainty (*u*$_a$) is reported with a coverage factor of k = 2. Disp. means dispersion, in %, between laboratory (lab) and reference (ref) $^{14}$C activity concentration (*a*) values.

| PT | Matrix | $a_{ref}$ | $u_{a_{ref}}$ | $a_{lab}$ | $u_{a_{lab}}$ | Disp. (%) | Z-Score |
|---|---|---|---|---|---|---|---|
| CSN, 2014 [13] | Leaves | 116 | 16 | 124 | 17 | 7 | 0.4 |
| CSN, 2015 [14] | Water | 32 | 6 | 38 | 3 | 17 | - |
| CSN, 2019 [15] | Water | 28 | 6 | 29 | 3 | 4 | 0.2 |
| Ref. Mat. | | | | | | | |
| IAEA-C-6 [12] | Oxalic acid | 31 | 3 | 35 | 4 | 13 | - |

These exercises serve to show the general capability of our methods. However, it is necessary to perform determinations closest to the detection limits of some of the procedures considered in this paper. Moreover, the ranges of activity concentrations shown via the historical analysis of the intercomparison providers studied in this work are higher than those required for some of our procedures. In the case of water samples, reference materials are not available.

In addition, for both solid and water samples, in order to validate a method, it is necessary to obtain more parameters (such as accuracy, precision, repeatability and linearity) than those evaluated in the PT (proficiency test) exercises. If reference materials are not available for method validation, as in our case, certain traced samples can be produced by the laboratory to obtain the quality parameters of the methods to be validated.

Thus, the objectives of this work are to describe and to apply the proposed activities to internally carry out a complete validation for $^{14}$C activity determination, according to the requirements for accreditation and developing strategies fit for purpose. This validation method would be applicable to the determination of other radionuclides for which there is a lack not only of intercomparison exercises but also of suitable reference materials—a situation that is becoming more and more frequent in the framework of dismantling and decommissioning (D and D) projects.

Therefore, traced sample preparation, measurement and analysis carried out for each $^{14}$C activity determination method is presented in this paper, having previously defined the quality parameters and their acceptance criteria, according to the performance evaluation of the ISO 13528 standard [16] and the Eurachem Laboratory Guide to Method Validation [17].

Currently, these methods are usually applied to real samples of water, air, food and crops; the results obtained being within the range of values published by the IRSN [2]. $^{14}$C activity concentration in water is below the detection limit ($L_D$) required (300 Bq m$^{-3}$), whereas in air it varies from 0.03 to 0.07 Bq m$^{-3}$, the $L_D$ required being 0.002 Bq m$^{-3}$. In food and crops, $^{14}$C activity concentration is in the range of 1 Bq wet kg$^{-1}$ ($L_D$ required) to more than 80 Bq wet kg$^{-1}$, depending on the food product and crop.

In order to ensure the quality of these results, analyses are usually performed by two different laboratories at least: the main laboratory and the control one, which performs between 5 and 10% of the analyses. This allows laboratories to support their validation results, as a field validation, and control their radiochemical methods.

## 2. Materials and Methods

### 2.1. Materials

In this section, the necessary materials to determine $^{14}$C activity in environmental samples via direct counting, bubbling and combustion methods are listed as follows:

—  Reagents of pro-analytical grade, which depend on the sample type and the analytical method to be applied: direct counting for water samples (when detection limits required > 2 Bq L$^{-1}$), bubbling and $CO_2$ absorption for water samples (when detection limits required < 2 Bq L$^{-1}$) and combustion for solid samples.
—  $^{14}$C standard solution in the form of $Na_2$$^{14}CO_3$ (from the Physikalisch-Technische Bundesanstalt (PTB, Braunschweig, Germany)), for direct and bubbling methods.
—  $^{14}$C standard solution in the form of toluene (SPEC–CHEC) (from PerkinElmer (Waltham, MA, USA), for the combustion method.
—  Carbo-Sorb® E (from PerkinElmer (USA)), for the bubbling and combustion methods.
—  Ultima Gold LLT scintillation cocktail (from PerkinElmer (USA)), for the direct method.
—  Permafluor® E$^+$ scintillation cocktail (from PerkinElmer (USA)), for the bubbling and combustion methods.
—  307 Sample Oxidizer (from PerkinElmer (USA)), for the combustion method.
—  Glass vials of 20 mL (from PerkinElmer (USA)).
—  An ultra-low background liquid scintillation spectrometer 1220 QUANTULUS$^{TM}$ (from PerkinElmer (USA)), which provides very low detection limits due to its active and passive shielding.

### 2.2. Methods

In this section, the strategy to enlarge the validation of the methods to determine $^{14}$C activity in environmental samples is described in three subsections: $^{14}$C activity determination methods, the preparation and measurement of samples and the analysis of results.

#### 2.2.1. $^{14}$C Activity Determination Methods

For all these methods, it is important to remark that before their treatment, water samples shall not be acidified to avoid $CO_2$ losses.

●  Direct counting for the determination of $^{14}$C activity via LSS

In the direct determination method [8], we mix 10 mL of water and 10 mL of Ultima Gold LLT scintillation cocktail in a 20 mL glass vial, prior to its measurement via LSS. Thus, the total $^{14}C$ is determined.

This method provides good detection efficiencies (around 65% in this work) and rapid and inexpensive results. Therefore, it is very useful for screening purposes.

However, it is limited by spectral interferers, i.e., potential radionuclides in water with a signal difficult to deconvolute from that of $^{14}C$ (e.g., $^{129}I$ with $E_{max}$ = 190.8 keV and $^{99}Tc$ with $E_{max}$ = 293.8 keV [1]). In addition, it is limited by variable quench, depending on water purity, and by the maximum volume of the sample that can be put into the vial (a certain number of mL), which leads to higher detection limits than those from other methods.

- Bubbling method for the determination of $^{14}C$ activity via LSS

The bubbling and $CO_2$ absorption method [8] also enables the measurement of total $^{14}C$. The application of this method starts with the addition of 150 mL of $H_2SO_4$ 1M to 100 mL of water. Then, the inorganic carbon in the form of bicarbonate or carbonate reacts with the sample, forming $CO_2$ gas, which is collected into 10 mL of Carbo-Sorb$^{\circledR}$ E and mixed with 10 mL of the Permafluor$^{\circledR}$ E$^+$ scintillation cocktail in a 20 mL glass vial, prior to its measurement with LSS. For determining $^{14}C$ in its organic form, an oxidation step should be included, so 4 mL of $KMnO_4$ 0.02M is also added to the sample. A scheme of the bubbling and $CO_2$ absorption system is shown in Figure 1.

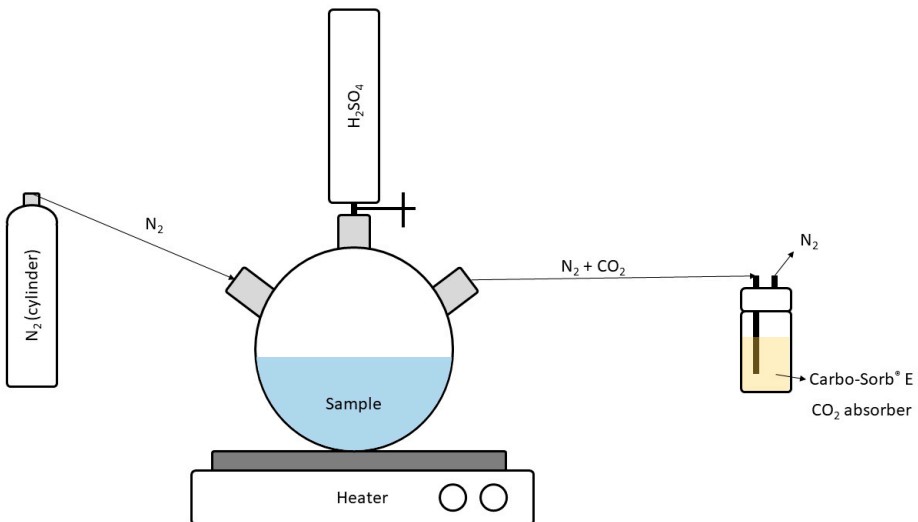

**Figure 1.** Scheme of the bubbling and $CO_2$ absorption system.

As this method is carbon selective, there are no spectral interferences during the sample measurement. Moreover, as it supports water samples greater than 50 mL [8], detection limits become lower than 2 Bq L$^{-1}$. However, the method is limited by the maximum quantity of the $CO_2$ absorber that is miscible with the scintillation cocktail, and it is strongly affected by its quench and chemiluminescence [9], which lead to low overall efficiencies (around 30% in this work).

Regarding air samples, $CO_2$ is retained on sodium hydroxide during sampling and then is precipitated as barium carbonate. After that, the precipitate is filtered, dried and treated as water samples via bubbling.

- Combustion method for determination of $^{14}C$ activity via LSS

The combustion method [18] is applied to solid samples. In this method, solid samples can be treated using a sample oxidizer (sketched in Figure 2), which combusts less than 1 mL of the sample material and separates $^{14}C$ in $CO_2$ form from $H_2O$ [18]. Then, $CO_2$ is retained in 10 mL of a $CO_2$ absorber (Carbo-Sorb$^{\circledR}$ E) and mixed with 10 mL of the liquid scintillation cocktail (Permafluor$^{\circledR}$ E$^+$) in a 20 mL glass vial, which is measured via LSS.

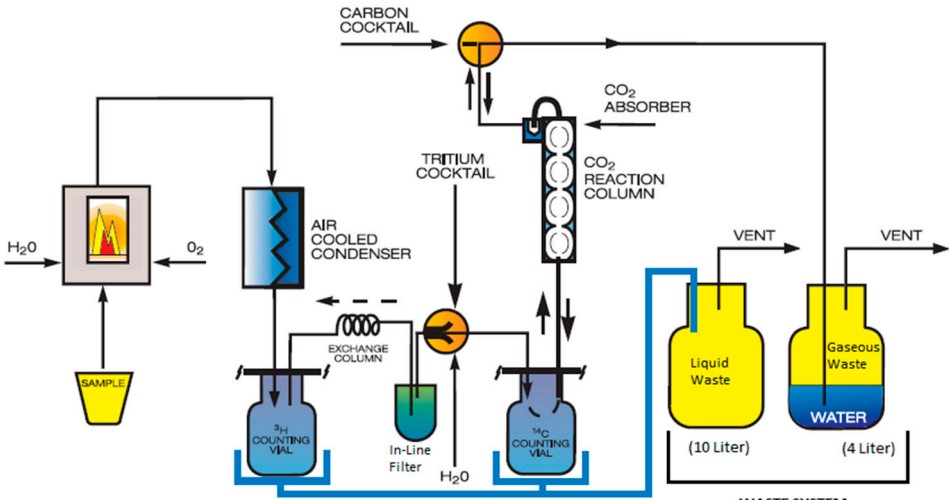

**Figure 2.** Sample oxidizer flow diagram [10].

This method is limited by the maximum quantity of sample that can be put into the sample oxidizer, which leads to long counting times to achieve low detection limits; in addition, the $CO_2$ absorber, which is a quench agent, leads to low overall efficiencies (around 40% in this work).

This method allows us to determine total $^{14}C$ in solid samples. However, spectral interferers, such as $^{137}Cs$, could affect the $^{14}C$ analysis, as $^{137}Cs$ is also partially retained in the $CO_2$ absorber when the sample contains high $^{137}Cs$ activities.

A possible solution to the spectral interference issue is the use of classical window settings with or without spill-up [19], defining two windows for two isotopes. Deconvolution could also be an interesting approach to solving the problem of a mix of spectral interferers [20], comparing the sample spectrum with a set of standard spectra. Both of these methods have shortcomings, and their use is not generic for all possible interferers. Neither of them has been used in this work.

### 2.2.2. Samples Preparation and Measurement

Considering the expected detection limit ($L_D$) for each of the three methods considered, a sample with a $^{14}C$ content between 1 and 10 times the expected $L_D$ and another sample with a $^{14}C$ content between 10 and 100 times $L_D$ were prepared for each method. Five aliquots of each one of these samples were taken to evaluate the following quality parameters: accuracy, precision, repeatability and, to some extent, linearity.

Two types of traced samples were prepared, water-based liquid and solid, the specificities of each preparation are given below.

To prepare water traced samples, we used distilled water and $^{14}C$ standard solution, from the Physikalisch-Technische Bundesanstalt (PTB), in the form of $Na_2{}^{14}CO_3$ because carbon is usually found as bicarbonate in environmental water samples [21].

For the combustion method, to be used for solid samples, we chose a soil matrix, because we consider it to be one of the most complex available solid materials and the one that naturally contains more radionuclides susceptible to producing interferences in the measurement. The $^{14}C$ standard solution used to trace it takes the form of toluene (SPEC–CHEC) from PerkinElmer. Although soil is a complex material that may contain both organic and inorganic carbon, the high temperature combustion process acts on both equally.

A soil sample, dried, ground and sieved to 0.5 mm, was soaked in the toluene solution and homogenised. Then, it was dried at room temperature and homogenised again, before the corresponding aliquots were taken to make the determinations.

When preparing soil samples, it was necessary to determine the $^{14}C$ activity concentration in the soil in advance. This activity was assessed applying the method to 5 samples

of 0.25 g and using counting times of 30 h both for samples and blanks (empty cellulose cones) measurements to obtain very low detection limits. In our case, $^{14}$C activity in the soil was always below the detection limit of 0.03 Bq g$^{-1}$.

The result of this determination was considered the background for traced soil samples, which were prepared using the same soil and in the same manner as the previous samples.

After preparation, each vial containing the prepared samples was stored for 6 h in the dark inside the spectrometer 1220 QUANTULUS$^{TM}$ and measured for 3 h using the high-energy measuring protocol ($^{14}$C). With this protocol, the $^{14}$C spectrum appears in the energy window defined between channels 100 and 450.

We calculated the activity concentration of each sample (*a*) using the following Equation (1):

$$a = \frac{r_g - r_0}{\varepsilon \cdot m} = (r_g - r_0) \cdot \omega \tag{1}$$

where *a* is the activity concentration of $^{14}$C; $r_g$ is the sample gross count rate; $r_0$ is the blank count rate; $\varepsilon$ is the detection efficiency in the direct method and the overall efficiency (the product of the chemical yield by detection efficiency) in the bubbling and combustion methods; *m* is the sample mass or volume; and $\omega$ is a parameter that summarises $(\varepsilon \cdot m)^{-1}$.

To calculate the detection efficiency, we prepared a calibration sample in the same way as routine samples but added a high and known amount of $^{14}$C tracer. After measuring, Equation (2) provides us the efficiency:

$$\varepsilon = \frac{r_s - r_g}{a_s \cdot V_s} \tag{2}$$

where $r_s$ is the gross count rate of the calibration sample; $a_s$ is the activity concentration of the $^{14}$C tracer added; and $V_s$ the volume of the $^{14}$C tracer.

2.2.3. Results Analysis

After sample preparation and measurement, the results should be analysed to assess the uncertainty, detection limit, accuracy, precision, repeatability and linearity of the methods to complete its validation.

For the evaluation of the uncertainties and detection limits, ISO/IEC Guide 98-3 [22] and ISO 11929 standard [23] and were followed, respectively.

For the other validation parameters, and among different possibilities, our strategy was to analyse the results as if they were the results obtained in the participation of a proficiency test exercise. Therefore, for the assessment of validation parameters of the ISO 13528 standard [16] and the Eurachem Laboratory Guide to Method Validation [17], recommendations were taken into account.

Then, the combined standard uncertainty of $^{14}$C activity concentration ($u_a$) was calculated using the following Equations (3)–(5):

$$u_a^2 = (r_g/t_g + r_0/t_0) \cdot \omega^2 + u_{rel}^2(\omega) \cdot a^2 \tag{3}$$

with:

$$u_{rel}^2(\omega) = u_{rel}^2(\varepsilon) + u_{rel}^2(m) \tag{4}$$

$$u_{rel}^2(\varepsilon) = u_{rel}^2(r_s - r_g) + u_{rel}^2(a_s) + u_{rel}^2(V_s) \tag{5}$$

where $t_g$ is the sample counting time and $t_0$ is the blank counting time.

From Equations (1)–(5), and taking the ISO 11929 standard [23] into account, expressions for decision threshold ($L_C$) and detection limit ($L_D$) were obtained:

$$L_C = k \cdot \omega \cdot \sqrt{r_0 \cdot (1/t_g + 1/t_0)} \tag{6}$$

$$L_D = \frac{(k^2 \cdot \omega)/t_g + 2 \cdot L_C}{1 - k^2 \cdot u_{rel}^2(\omega)} \tag{7}$$

where $k$ is the quantile of the standard normal probability distribution with a value of 1.65 for a confidence level of 95%. In this case, the probabilities 1-$\alpha$ and 1-$\beta$ from the definitions of the decision threshold ($L_C$) and the detection limit ($L_D$) [23] were taken as equal.

According to the ISO 5725-1 standard [24], the accuracy of a method refers to the closeness of the agreement between the test result and an accepted reference value. There are different expressions for calculating the accuracy of a method, the main difference between them being the uncertainty consideration. In this work, accuracy was calculated as the $u_{test}$ parameter [16], which considers uncertainties, following Equation (8):

$$u_{test} = \frac{\left| a_{ref} - \bar{a} \right|}{\sqrt{u_{a_{ref}}^2 + u_{\bar{a}}^2}} \tag{8}$$

where $a_{ref}$ is the reference activity concentration in the samples; $\bar{a}$ is the mean value of measured activity concentration in a set of samples or replicates; and $u_{a_{ref}}^2$ and $u_{\bar{a}}^2$ are the standard uncertainties of both activities.

The acceptance criterion taken for accuracy is $u_{test} \leq 2.58$, for a confidence level of 99% [16].

According to the ISO 5725-1 standard [24], the precision of a method refers to the degree of agreement between the independent results of its test, obtained under repeatable or reproducible conditions. Repeatability conditions can be accomplished in a short period by a laboratory with the same reagents, equipment and staff; however, reproducibility conditions can be achieved between different laboratories or in different periods by a laboratory with the same reagents, equipment and staff.

As with the accuracy, there are different expressions for calculating the precision of a method, the main difference between them being the uncertainty consideration. In this work, precision was calculated under repeatability conditions as the $P$ parameter [16], following Equation (9):

$$P = \sqrt{\left( u_{a_{ref}}/a_{ref} \right)^2 + \left( u_{\bar{a}}/\bar{a} \right)^2} \cdot 100 \tag{9}$$

Considering the expected performance of these methods, the experience of the laboratory and the needs of our clients, the acceptance criterion that we considered adequate for precision was $P \leq 10$ %, for both levels of activity, i.e., 1–10 $L_D$ and 10–100 $L_D$.

Regarding the repeatability of a method, it is usually expressed as the relative standard deviation ($RSD$) of a set of at least three samples or replicates and is calculated according to Equations (10) and (11):

$$RSD = {}^{SD}/_{\bar{a}} \cdot 100 \tag{10}$$

with:

$$SD = \sqrt{\left( \frac{\sum (a_i - \bar{a})^2}{n - 1} \right)} \tag{11}$$

where $RSD$ is the relative standard deviation, in %; $SD$ is the standard deviation; $a_i$ is the activity concentration of each sample or replicate; and $n$ is the number of samples or replicates to test.

In this case, the acceptance criterion is $RSD \leq 10\%$, from our laboratory experience.

Finally, with regard to the linearity of our methods, this was evaluated at two different activity concentration levels. The signal provided by the measurement equipment is intrinsically linear, its efficiency being independent of the activity of the sample measured. In the sample preparation, there were also no linearity problems with the sample activity, since there are no radiochemical separations were involved. Therefore, we can assume that

performing determinations at two different activities, one close to the limit of detection and the other at least one order of magnitude higher, is sufficient to prove that our methods work correctly within the range of activities that match values usually found in routinely analysed samples.

## 3. Results and Discussion

### 3.1. Direct Counting for the Determination of $^{14}$C Activity via LSS

Following the methods described in Section 2.2., the measurement results of the set of water samples spiked with $^{14}$C standard solution and analysed according to the ISO 13162:2021 standard [8], namely, via direct counting, are summarised in Table 2.

**Table 2.** $^{14}$C activity concentration (*a*), in Bq L$^{-1}$, in water samples spiked with $^{14}$C standard solution, using the direct method. Uncertainty ($u_a$) is reported with a coverage factor of k = 1.

| | | **Low *a*** | | | **High *a*** | |
| | *a* (Bq L$^{-1}$) | $u_a$ (Bq L$^{-1}$) | $L_D$ (Bq L$^{-1}$) | *a* (Bq L$^{-1}$) | $u_a$ (Bq L$^{-1}$) | $L_D$ (Bq L$^{-1}$) |
|---|---|---|---|---|---|---|
| Reference value ($a_{ref}$) | 13.2 | 0.3 | - | 132 | 3 | - |
| Sample 1 | 11.3 | 0.8 | 2.0 | 131 | 3 | 2 |
| Sample 2 | 11.9 | 0.8 | 2.0 | 134 | 3 | 2 |
| Sample 3 | 11.3 | 0.8 | 2.0 | 135 | 3 | 2 |
| Sample 4 | 11.0 | 0.8 | 2.0 | 131 | 3 | 2 |
| Sample 5 | 11.7 | 0.8 | 2.0 | 130 | 3 | 2 |

As we can see in the table, at each activity concentration, reference and measured activity concentrations are very close to each other. The relative standard uncertainty of measured activity concentration is below 7% at low activity level and 2% at high activity level.

The results of the analysis of this set of samples are presented in Table 3.

**Table 3.** Summary of the quality parameters analysis for the direct method, with water samples spiked with $^{14}$C standard solution.

| | **Low *a*** | **High *a*** |
|---|---|---|
| Number of samples | 5 | 5 |
| Reference *a* value, $a_{ref}$ (Bq L$^{-1}$) | 13.2 ± 0.3 | 132 ± 3 |
| Measured mean *a*, $\bar{a}$ (Bq L$^{-1}$) | 11.4 ± 0.8 | 132 ± 3 |
| Accuracy, ($u_{test}$) | 2.06 | 0.03 |
| Precision, *P* (%) | 7.35 | 3.13 |
| Repeatability, *RSD* (%) | 3.13 | 1.86 |

This table shows that all the studied parameters are in the acceptance range at both low and high activity levels: accuracy as $u_{test}$ is below 2.58 and precision and repeatability are below 10%. Therefore, and also taking into account the results of the intercomparison exercises in which we have participated, the method can be considered validated for water samples.

Despite some limitations, the results are acceptable, so the detection limit and uncertainty of the direct method can be defined as 2 Bq L$^{-1}$ and 2–7%, respectively, for 10 mL of water, using 100 min for sample counting and 300 min for blank counting. The detection efficiency of the method is around 65% and linearity has been proven in an activity concentration range from 13 to 130 Bq L$^{-1}$.

### 3.2. Bubbling Method for the Determination of $^{14}$C Activity via LSS

According to the methods described in Section 2.2, measurement results in the set of water samples spiked with $^{14}$C standard solution and prepared via bubbling are presented in the following Table 4.

**Table 4.** $^{14}$C activity concentration ($a$), in Bq L$^{-1}$, in water samples spiked with $^{14}$C standard solution, via the bubbling method. Uncertainty ($u_a$) is reported with a coverage factor of k = 1.

| | **Low $a$** | | | **High $a$** | | |
| --- | --- | --- | --- | --- | --- | --- |
| | $a$ (Bq L$^{-1}$) | $u_a$ (Bq L$^{-1}$) | $L_D$ (Bq L$^{-1}$) | $a$ (Bq L$^{-1}$) | $u_a$ (Bq L$^{-1}$) | $L_D$ (Bq L$^{-1}$) |
| Reference value ($a_{ref}$) | 0.89 | 0.02 | - | 8.8 | 0.2 | - |
| Sample 1 | 0.91 | 0.09 | 0.14 | 8.2 | 0.7 | 0.1 |
| Sample 2 | 0.92 | 0.09 | 0.14 | 9.2 | 0.7 | 0.1 |
| Sample 3 | 0.91 | 0.09 | 0.14 | 8.9 | 0.7 | 0.1 |
| Sample 4 | 0.86 | 0.09 | 0.14 | 8.9 | 0.7 | 0.1 |
| Sample 5 | 0.94 | 0.09 | 0.14 | 9.3 | 0.7 | 0.1 |

As seen in the table, the relative standard uncertainty of measured activity concentration is below 10% at both low and high activity levels. Moreover, reference and measured activity concentrations are very similar, at each activity concentration, with the relative standard deviation of the results being below 5%.

The results of the analysis of this set of samples are presented in Table 5.

**Table 5.** Summary of the quality parameters analysis for the bubbling method, with water samples spiked with $^{14}$C standard solution.

| | **Low $a$** | **High $a$** |
| --- | --- | --- |
| Number of samples | 5 | 5 |
| Reference $a$ value, $a_{ref}$ (Bq L$^{-1}$) | 0.89 ± 0.02 | 8.8 ± 0.2 |
| Measured mean $a$, $\bar{a}$ (Bq L$^{-1}$) | 0.91 ± 0.09 | 8.9 ± 0.7 |
| Accuracy ($u_{test}$) | 0.20 | 0.14 |
| Precision, $P$ (%) | 10.2 | 8.19 |
| Repeatability, $RSD$ (%) | 3.25 | 4.83 |

This table shows that in the bubbling method for air and water samples, with low or high activity concentration, we also achieved good results regarding accuracy (<2.58), precision (<15%) and repeatability (<15%). Therefore, and also taking the results of the intercomparison exercises in which we have participated into account, the method can be considered validated for water samples.

Using this method, the sample volume can be arbitrarily large, although we routinely use 100 mL. This means that, even if this method consumes more time and resources than the previous method, detection limits can be orders of magnitude lower, making it more suitable when very low detection limits are to be achieved.

In view of the results obtained, the detection limit and the uncertainty of the bubbling method can be defined as 0.14 Bq L$^{-1}$ and 5–10%, respectively, for 100 mL of water with 5 h of sample counting time and 30 h of blank counting time. The overall efficiency of the method is around 32% and linearity has been proven with activity concentrations between 0.9 to 9 Bq L$^{-1}$.

### 3.3. Combustion Method for Determination of $^{14}$C Activity via LSS

Finally, measurement results of the set of soil samples spiked with $^{14}$C standard solution and prepared using a sample oxidizer are summarised in Table 6.

**Table 6.** $^{14}$C activity concentration ($a$), in Bq g$^{-1}$, in soil samples spiked with $^{14}$C standard solution, using the combustion method. Uncertainty ($u_a$) is reported with a coverage factor of k = 1.

| | Low $a$ | | | High $a$ | | |
|---|---|---|---|---|---|---|
| | $a$ (Bq L$^{-1}$) | $u_a$ (Bq L$^{-1}$) | $L_D$ (Bq L$^{-1}$) | $a$ (Bq L$^{-1}$) | $u_a$ (Bq L$^{-1}$) | $L_D$ (Bq L$^{-1}$) |
| Reference value ($a_{ref}$) | 0.50 | 0.01 | - | 28.7 | 0.6 | - |
| Sample 1 | 0.56 | 0.03 | 0.07 | 28.6 | 0.7 | 0.1 |
| Sample 2 | 0.50 | 0.03 | 0.07 | 28.2 | 0.7 | 0.1 |
| Sample 3 | 0.52 | 0.03 | 0.07 | 28.8 | 0.7 | 0.1 |
| Sample 4 | 0.54 | 0.03 | 0.07 | 27.8 | 0.7 | 0.1 |
| Sample 5 | - | - | - | 28.8 | 0.7 | 0.1 |

As we can see in the above table, after the combustion of soil samples with organic $^{14}$C standard solution, reference and measured activity concentrations appear to be very close to each other, at either low or high activity concentrations. In this case, the relative uncertainty of measured activity concentration is below 6% at both low and high activity levels and the relative standard deviation of the results is below 5%.

It should be clarified that the absence of a result in the low activity concentration set is due to an outlier.

The results of the analysis of this set of samples are presented in Table 7.

**Table 7.** Summary of the quality parameters analysis for the combustion method, with soil samples spiked with $^{14}$C standard solution.

| | Low $a$ | High $a$ |
|---|---|---|
| Number of samples | 4 | 5 |
| Reference $a$ value, $a_{ref}$ (Bq g$^{-1}$) | 0.50 ± 0.01 | 28.7 ± 0.6 |
| Measured mean $a$, $\bar{a}$ (Bq g$^{-1}$) | 0.53 ± 0.03 | 28.4 ± 0.7 |
| Accuracy ($u_{test}$) | 0.95 | 0.33 |
| Precision, $P$ (%) | 6.00 | 3.23 |
| Repeatability, $RSD$ (%) | 4.87 | 1.52 |

This table shows that in the combustion method for solid samples, with low or high activity concentration, we also achieved good results regarding accuracy (<2.58), precision (<15%) and repeatability (<15%). So, linearity was also proven.

Obtained results are acceptable, so the method can be considered validated in the interim, pending certified reference materials or intercomparison exercises. Its detection limit and uncertainty can be defined as 70 Bq kg$^{-1}$ and 2–6%, respectively, for 0.25 g of solid sample, using 3 h for sample counting and 30 h for blank counting. The overall efficiency of the method is around 40% and linearity has been proven with activity concentrations of between 500 to 29,000 Bq kg$^{-1}$.

In this method, the volume of the container of the combustion equipment limits the size of the sample. In the equipment employed in this work, this volume is 1 mL; the amount of sample in grams, and therefore, the detection limit in Bq g$^{-1}$, will depend on the bulk density of the sample, being lower for soil samples and higher for organic samples.

### 3.4. Field Validation in Environmental Samples

After validating the methods described in Section 2.2, $^{14}$C activity concentration has been determined in a set of air, water, soil, sediments, food and biota samples. Most of these belong to the Spanish National Radiological Surveillance Network for the environmental radiological monitoring of nuclear power plants and nuclear fuel cycle facilities.

In order to ensure the quality of the results obtained within the framework of the Radiological Surveillance Network, analyses are performed by two different laboratories at least: a main laboratory and a control one, which performs between 5 and 10% of the

analyses. The Laboratory of Low Activity Measurements (LMBA) of the University of the Basque Country (UPV/EHU) is sometimes the main laboratory and sometimes the control one.

This allows our laboratory to carry out $^{14}$C field validation through radioanalytical methods in environmental samples, which supports internal laboratory validation and external validation through proficiency tests and interlaboratory comparisons.

In Table 8, environmental samples utilized for the field validation and obtained relevant results are summarised.

**Table 8.** Overall efficiency, in %, activity concentration uncertainty ($u_a$), in %, with a coverage factor of k = 1, and detection limit ($L_D$), in Bq L$^{-1}$ or Bq kg$^{-1}$, for a set of different matrix samples analysed via the direct counting, bubbling or combustion methods. For the overall efficiency and $L_D$, mean and standard deviation values are considered. $L_D$ for air and water is given in Bq L$^{-1}$ and for solid samples is given in Bq kg$^{-1}$.

| Matrix | Number of Samples | Method | Overall Efficiency (%) | $u_a$ (%) | $L_D$ (Bq L$^{-1}$ or kg$^{-1}$) |
|---|---|---|---|---|---|
| Water | 550 | Direct counting | 62–66 | <7 | 0.26–0.30 |
| Water | 600 | Bubbling | 30–42 | <10 | 0.12–0.16 |
| Air | 52 | Bubbling | 31–47 | <10 | $1.6–2.0 \times 10^{-6}$ |
| Soil and sediments | 4 | Combustion | 32–36 | <6 | 8.5–9.5 |
| Food | 210 | Combustion | 38–54 | <6 | 6–10 |
| Biota | 16 | Combustion | 37–53 | <6 | 6–10 |

As we can see in the table, a significant number of samples have been analysed using the $^{14}$C radioanalytical methods: 550 samples via direct counting, 652 samples via bubbling and 230 samples via combustion (i.e., 1432 samples in total).

All the results obtained are in accordance with those from the main or the control laboratory. Moreover, obtained efficiency and activity concentration relative uncertainty are very similar to those from the internal validation:

− Direct counting: around 65% of detection efficiency and 2–7% of uncertainty.
− Bubbling method: around 32% of overall efficiency and 5–10% of uncertainty.
− Combustion method: around 40% of overall efficiency and 2–6% of uncertainty.

However, detection limits are lower than the ones obtained in the internal validation, in order to achieve surveillance requirements. To lower them, we increased only the counting time.

## 4. Conclusions

$^{14}$C is a meaningful radionuclide for the assessment of the impacts of nuclear activity or the fossil fuel industry on the environment. However, there are still many challenges in these procedures for determining $^{14}$C activity in environmental samples via liquid scintillation spectrometry (LSS).

Regarding the development and adaptation of the methods, we should take into account that the direct and the combustion methods described in this work must deal with spectral interference issues (beta-emitters with energy spectra overlapping those of $^{14}$C) and are limited by the small sample volumes (10 mL and 1 mL, respectively) that can be utilized in each method.

To solve spectral interferer issues, the bubbling method is an adequate approach, as it is more selective than the direct method but is limited to water and air samples.

With regard to the accreditation of the methods for $^{14}$C activity determination in environmental samples via LSS under the ISO/IEC 17025 [11], the lack of reference materials and the few and limited proficiency tests and interlaboratory comparisons lead to laboratories having to design their own internal validation strategies and to prepare fit for purpose materials.

In this work, the validation strategy for three different radioanalytical methods to determine [14]C activity in environmental samples via LSS—the direct counting and bubbling methods, both based on the ISO 13162:2021 standard, for water samples, and the combustion of solid samples using a sample oxidizer—has been described and analysed.

In all the methods, the studied quality parameters (accuracy, precision and repeatability) are within the acceptance range at two activity levels (around 10 to 100 times the detection limits), which enabled us to assess their detection limits, uncertainties and linearity, despite their limitations.

In the direct method for water samples, the detection limit obtained was 20 mBq for 10 mL of water; in the bubbling method for water and air samples the detection limit was 14 mBq for 100 mL of water or 2.6 m$^3$ of air; and in the combustion method for solid samples the detection limit was 17.5 mBq when using a sample of 0.25 g.

In all the methods, relative uncertainty was below 10% and linearity was proven with a rather large range of activity concentrations (from around 5 to 100 times the detection limit), within those found in environmental samples. Regarding overall method efficiencies, the highest was that of the direct method (65%), as there was no chemical procedure and the water sample was simply mixed with the scintillation cocktail, whereas the lowest was obtained via the bubbling method (32%). Finally, regarding counting times, to achieve the detection limits found, the shortest time was required for the direct counting method (100 min), whereas longer times were required for the bubbling and combustion methods (5 and 3 h, respectively).

These outcomes are supported by a field validation conducted within the framework of the Spanish National Radiological Surveillance Network, as the results of each [14]C radioanalytical method are in accordance with the ones from the other laboratory.

In conclusion, in spite of all the aforementioned difficulties, our analytical procedures for [14]C activity determination via LSS have been validated and accredited under ISO/IEC 17025 [11].

**Author Contributions:** Conceptualization, M.H.; methodology, S.R., R.I., M.T.R. and M.H.; software, S.R. and R.I.; validation, S.R., R.I., M.T.R. and M.H.; formal analysis, S.R., R.I. and M.H.; investigation, S.R., R.I., M.T.R. and M.H.; resources, R.I. and M.H.; data curation, S.R., R.I. and M.H.; writing—original draft preparation, S.R.; writing—review and editing, S.R. and M.H.; visualization, S.R.; supervision, M.H.; project administration, R.I. and M.H. All authors have read and agreed to the published version of the manuscript.

**Funding:** We would like to acknowledge the open access funding provided by the University of the Basque Country (UPV/EHU).

**Institutional Review Board Statement:** Not applicable.

**Data Availability Statement:** Data supporting reported results can be found in the CSN (Spanish Nuclear Safety Council) virtual map of radiological environmental surveillance in Spain, via the following link: https://www.csn.es/kprgisweb2/index.html?lang=en (accessed on 23 February 2023).

**Conflicts of Interest:** The authors declare no conflict of interest.

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
