# Peer review of "Quality Assurance of Potential Radioanalytical Methods for 14C in Environmental Samples"

_environments, doi:10.3390/environments10060098_

Round 1

Reviewer 1 Report

I believe this review has scientific merit and it is very useful topic. The subject of the manuscript “Quality assurance of 14C measurements in environmental samples” is in good relevance with the scope of IJERPH however, minor revision is needed. After analyzing the manuscript. I have some suggestions for the authors listed below before being considered for publication.

1-      I believe a more sounding title could be used such as "Validation of Potential Radioanalytical Methods into the Quality assurance of 14C measurements in environmental samples” or similar

2-      In the abstract, I would love seeing a sentence showing the importance of 14C measurement

3-      The abstract needs further elaboration in the methods explaining how the study was conducted.

4-      In the introduction, it is essential to define abbreviations at the first usage such as IAEA, NPL, IARMA, IRSN and CSN.

5-      In materials and methods, it is preferably that you have a reference for the methods followed.

6-      In the Materials and Methods section, I recommend to add an image or sketch for the setup and samples. 

7-      In materials, provide the (manufacture, country) for the 1220 QUANTULUSTM2 LSS

8-      In the subheading and in the whole manuscript, try to replace “by LSS” with “by the LSS”

9-      In the Conclusions, line 460, no need to write liquid scintillation spectrometry (LSS) in full use LSS

Reviewer 2 Report

This paper can be recommended for publication after a more detailed disclosure of some ambiguities and uncertainties.

1.  Please advise what other isotopes may cause interference with 14C?

2.  The following paragraph (lines 67-71) needs supporting references.

3. Is it necessary to indicate the measurement error in the Tables 1 and 2?

Reviewer 3 Report

Dear Authors

The purpose of your many years' of research, is to become accredited under ISO/IEC 17025. You have reached that goal. Congratulation.

Unfortunately, your paper is very hard to read. You assume that all magazine readers are familiar with the field of radionuclide measurement and characterization. So you talk about the results of proficiency tests CSN 2014, CSN 2015, to 2019 without any introduction to what CSN is and what the CSN acronym means. You are introducing also acronyms NPL, IARMA, IRSM, INSIDER 2020, etc., with no explanation. This is very confusing. Also, IAEA is not a well-known abbreviation but is also an acronym, and the readers deserve a proper introduction. When reading your paper, I expected that you have developed a new measurement method, but I could not find any new method, only refined procedures, and recipes on how to solve the problems when there is missing reference data or other problems. I appreciate the hard work you have done to collect enough positive results. I suggest you edit the text to highlight your novel achievements, make the paper more interesting, and avoid the boring report-type style.

Dear Authors, I wish you a good luck.

Reviewer 4 Report

The authors show an interesting contribution entitled "Quality Assurance of 14C Measurements in Environmental Samples", in which three different methodologies were used to estimate the activation concentration in C-14 samples. However, there are some considerations that authors should keep in mind:

Authors should exercise extra caution regarding significant figures used to express uncertainty and ensure that rounding is done evenly. Table 1, for example, uses two significant figures, while Table 3 uses only one. Authors should aim to standardize the number of significant figures used in all tables and figures in their contribution.

 Results related to direct counting for determination of 14C activity by LSS for low activity, as depicted in Tables 2 and 3, exhibited activity values that were inferior to those of the reference. Additionally, the aforementioned values failed to intersect. Thus, authors to should justified better this observed outcome.

Reviewer 5 Report

The manuscript "Quality assurance of 14C measurements in environmental samples" discusses the measurement of 14C activity using LSS. While the data and discussion contained in the manuscript are useful for researchers studying radiocarbon as a tracer, it is apparent that the author did not fully present and discuss their findings. Therefore, it is recommended that the manuscript should be revised and submitted in an improved version.

The authors conducted a research on the validation of radiocarbon measurement in various samples.

However, there are some shorcomins in overall description, and there is no novelty in the results (i.e., method validation by CRM prior to measurment is not new).

Therefore, it seems that the description needs to be improved in future submissions, and it is recommended to focus on what they have discovered in the results.

Round 2

Reviewer 3 Report

Dear authors

The paper is improved to an acceptable level.

Congratulation

Author Response

Dear reviewer,

Thank you very much for your comment.

Reviewer 5 Report

Dear Authors,

The revised manuscript does seem to be much improved compared to the previous version. However, the originality of this paper was still not obvious. In addition, a paper cannot be based solely on a validated measurement method. Thus, it seems necessary to add field validation experiments using such measurement methods.

Author Response

Dear reviewer,

We agree with your comment.

Regarding field validation experiments, we have added a new subsection (3.4) in the results and discussion section (3), which contains a summary of 14C analyses performed in environmental samples from the National Radiological Surveillance Network. These analyses allow our laboratory to control 14C radiochemical methods and support their internal and external validations.

In this sense, we have also added the main points of this new subsection to the abstract, introduction and conclusions.

Regarding the originality of the paper, we have added some lines to the introduction in order to enhance the challenge of validating radioanalytical methods.

The changes to the previous version of our paper are highlighted in green in the new version to ease the review process.